# Crop Identification Using Deep Learning on LUCAS Crop Cover Photos

**DOI:** 10.3390/s23146298

**Published:** 2023-07-11

**Authors:** Momchil Yordanov, Raphaël d’Andrimont, Laura Martinez-Sanchez, Guido Lemoine, Dominique Fasbender, Marijn van der Velde

**Affiliations:** 1SEIDOR Consulting S.L., 08500 Barcelona, Spain; 2European Commission, Joint Research Centre (JRC), 21027 Ispra, Italy; raphael.dandrimont@ec.europa.eu (R.d.); laura.martinez-sanchez@ec.europa.eu (L.M.-S.); guido.lemoine@ec.europa.eu (G.L.); d.fasbender@iweps.be (D.F.); 3Walloon Institute of Evaluation, Foresight and Statistics (IWEPS), 5001 Namur, Belgium

**Keywords:** plant recognition, agriculture, computer vision, deep learning, data valorization, mapping from imagery, image classification algorithms

## Abstract

Massive and high-quality in situ data are essential for Earth-observation-based agricultural monitoring. However, field surveying requires considerable organizational effort and money. Using computer vision to recognize crop types on geo-tagged photos could be a game changer allowing for the provision of timely and accurate crop-specific information. This study presents the first use of the largest multi-year set of labelled close-up in situ photos systematically collected across the European Union from the Land Use Cover Area frame Survey (LUCAS). Benefiting from this unique in situ dataset, this study aims to benchmark and test computer vision models to recognize major crops on close-up photos statistically distributed spatially and through time between 2006 and 2018 in a practical agricultural policy relevant context. The methodology makes use of crop calendars from various sources to ascertain the mature stage of the crop, of an extensive paradigm for the hyper-parameterization of MobileNet from random parameter initialization, and of various techniques from information theory in order to carry out more accurate post-processing filtering on results. The work has produced a dataset of 169,460 images of mature crops for the 12 classes, out of which 15,876 were manually selected as representing a clean sample without any foreign objects or unfavorable conditions. The best-performing model achieved a macro F1 (M-F1) of 0.75 on an imbalanced test dataset of 8642 photos. Using metrics from information theory, namely the equivalence reference probability, resulted in an increase of 6%. The most unfavorable conditions for taking such images, across all crop classes, were found to be too early or late in the season. The proposed methodology shows the possibility of using minimal auxiliary data outside the images themselves in order to achieve an M-F1 of 0.82 for labelling between 12 major European crops.

## 1. Introduction

The deep learning (DL) paradigm is regarded as the gold standard of the machine learning (ML) community [1]. While the trade-off between the better performance of a model and the amount of data and resources necessary is still present, there is clearly a significant improvement with DL methods, especially so in image classification tasks. Recent advancements in convolutional neural networks (CNNs) have made popular classification tasks ever more affordable.

In an operational context, the ability to perform on-device processing provides an option to anyone wanting to implement the technology to keep computational overhead low. A leading architecture in this regard is MobileNet [2] and its third generation flavours [3]. MobileNets are convenient, as they perform on par with other state-of-the-art architectures on popular benchmarking datasets, but have up to 20 million fewer parameters.

Another point in making DL models operational is the proper use of post-processing techniques, meaning any manipulations executed on the model’s output data. In an image classification setting, this refers to operations executed on the data after the output of the softmax activation function. Popular post-processing approaches include combining random forest or support vector machine classifiers after the CNN output, majority voting [4], patch aggregation [5], and thresholding. The latter is a popular choice, as it is simple to implement; one must keep only the examples, for which the network has a maximum probability (MP) for the winning class higher than the threshold. Recent developments try to re-map the base probabilistic output to a metric of higher versatility, taking notions from information theories such as Shannon information and entropy [6].

The agricultural sector is adopting these technological developments [7], with DL-aided computer vision (CV) in particular being crucial for robotic tasks such as the inspection, evaluation, and execution of management interventions [8]. Ultimately, these innovations should decrease costs and increase the resource use efficiency and the precision of food production systems. This may relate to the certification of management practices [9], the traceability of products [10], communications towards consumers [11], and activities in the realm of citizen science and food related topics [12], including biodiversity [13]. In technical terms, the possibilities have already been successfully tested for weed management [14], crop disease recognition and management [15], and harvesting operations [16].

Activities also focus on training data collection and curation for increasingly specific applications, such as precision agriculture in field conditions [17,18] and robotic CV control [19]. In the Earth observation (EO) domain, datasets such as CropHarvest [20], which has more than 90,000 worldwide geographically diverse samples with labels, and the LUCAS 2018 Copernicus polygons [21], which have almost 60,000 stratified samples in the European Union (EU), demonstrate the push from the community to have open and free data to facilitate ML- and DL-driven research. In this manuscript, we focus on recognizing crops on a selection of legacy close-up photos from the five tri-annual Land Use/Cover Area frame Survey (LUCAS) surveys from 2006 to 2018 in the EU [22].

A fitting application in the EU context is the ability of technology to deliver to the needs of regulating bodies that administer the technical regulations of the EU’s Common Agricultural Policy (CAP). The CAP is the largest item on the EU budget, amounting to EUR 58.38 billion in 2022, including funds allocated for rural development, market measures, and income support (European Commission, The common agricultural policy at a glance, https://ec.europa.eu/info/food-farming-fisheries/key-policies/common-agricultural-policy/cap-glance_en, accessed on 17 May 2023). Thus, developing technology for automatizing the application process and the provision of evidence for practices required under subsidy schemes is in demand by paying agencies of the member states (MS).

While Copernicus-Sentinel-based monitoring of the CAP area subsidies is being developed and implemented [23], ground-based information in the form of geo-tagged pictures [24] can support and complement the checks by monitoring approach (CbM). CbM relies on Copernicus application-ready data (CARD), in conjunction with geospatial information from the land parcel identification systems (LPIS) and geo-spatial aid applications (GSAA) to provide wall-to-wall coverage of EU territory by extracting parcel-level information of markers of specific practices (see Devos et al. [23]). In situations in which Sentinel-based checks do not lead to conclusive results, geo-tagged pictures can be used to support and complement checks. Such processing chains may have to be developed for each specific agri-environmental practice for which evidence is needed. In the current CAP programming period (2023–2027), this includes practices under GAEC (Good Agricultural and Environmental Conditions) conditionality, as well as eco-schemes and agri-environmental and climate measures.

### Objectives

The aim of the present research is to benchmark and test computer vision models to recognize major and mature European crops (MMEC) on close-up photos in a practical agricultural-policy-relevant context. Specific objectives are:To select and publish a subset of LUCAS cover photos representative of major and mature crops across the EU for training purposes.To deploy and benchmark a set of MobileNet computer vision models to recognize crops on close-up pictures and identify the best-performing model.To explore the use of probability- and entropy-based metrics to threshold and filter correct and incorrect classifications.To illustrate the applications and limitations of the model for inference in a practical and agricultural-policy-relevant context.

## 2. Materials and Methods

The methodological approach in the manuscript consists of (1) the procedure to select close-up LUCAS cover MMEC photos; (2) the training, validating, and testing of a large set of MobileNet-based computer vision models; (3) applying the best model to inference photos across the EU; (4) evaluating model performance using metrics derived from information theory to filter and understand why photos are not classified well; (5) testing model performance against images exhibiting a series of unfavorable/out-of-scope conditions; (6) illustrating practical implications for protocol development. More specifically, the workflow is presented in Figure 1.

### 2.1. Data

#### 2.1.1. LUCAS Cover Photos

LUCAS Cover has been a part of the core LUCAS survey since its inception, and accordingly, data have been collected for all five campaigns form 2006 to 2018. A total of 875,661 LUCAS Cover photos have been collected, and 874,646 of those were published after anonymization and curation [22]. In contrast to other types of LUCAS core imagery (four N, S, W, E, photos in the cardinal directions, and the point photo P), the Cover (C) photos, by protocol, must show the cover on the ground at the GPS location where the survey is carried out in such a way that the relevant crop or plant can easily be identified during data quality control operations. An example of one photo per selected crop is shown in Figure 2. The selection was made with reference to the main crops that are monitored and forecast by the European Commission’s Joint Research Centre crop forecasting activities (AGRI4CAST, formerly MARS; see van der Velde et al. [25]). By omitting some classes due to data insufficiency and including temporary grassland, the number of crops became 12. These are: common wheat (B11), durum wheat (B12), barley (B13), rye (B14), oats (B15), maize (B16), potatoes (B21), sugar beet (B22), sunflower (B31), rape and turnip rape (B32), soya (B33), and temporary grassland (B55). The LUCAS Cover dataset is one of the two main inputs to the study, as shown on the left in Figure 1.

#### 2.1.2. Crop Calendars and Harmonization

One of the objectives of the study is the identification of mature crops on geo-tagged LUCAS imagery. The rationale for this is that, from an operational standpoint, the mature stage of the crop is the one in which it is most recognizable. The mature stages of the selected crops have to be firstly ascertained. One way of doing so is by collecting all crop calendars from the variety of sources available, harmonizing them into a common format, extracting the harvest period for each crop, and finally, through the use of expert knowledge, derive the pre-harvest mature stage of the crop.

A crop calendar (CC) is a schedule that provides timely information about crops in their respective agro-ecological zones. They are usually provided in tabular or gridded form and cover the space of a calendar year by dividing it into the planting, vegetative, and harvest stages of the respective crop. For the present purposes, CCs were gathered from various sources (Table A1) and harmonized to a common style (AGRI4CAST) that hosts the data in tabular and numeric format, facilitating further processing. Because of a lack of alternative data sources, the intended use of CCs was not taken into consideration (see Section 4.3). From a processing standpoint, certain steps had to be taken to account for instances where more than one variety (spring/winter, or early/late ware varieties) of the same crop was cultivated in a country. The decision was made to exclude countries that cultivate both varieties and use the CC information for only those countries that cultivate the winter and early ware variety, with the information for the excluded countries being populated by expert knowledge.

#### 2.1.3. Expert Knowledge Gap Filling and Mature Pre-Harvest Stages

After harmonizing the CCs and extracting harvest stages at a national level, this study filled the gaps and validated the result by means of expert knowledge. One way of identifying gaps is using the information from the JRC MARS bulletins [26]. These bulletins offer information on crop growth conditions and yield forecast at the EU level and for neighbouring countries, such as the UK, Ukraine, Black Sea area, and Maghreb. The rationale here is that if there is information in the bulletin about the yield of a certain crop for a certain country, then the crop is obviously cultivated in the respective country, and, ergo, CC information about it should be present. After identifying the gaps, they were filled with all available information, which was comprised of interpolations from the COST 725 phenology network [27], and expert knowledge. A breakdown of all the information gathered and the sources it was collected from is available in Appendix A, Figure A1. The final step was acquiring the mature, pre-harvest conditions of the crops. This was accomplished again with the use of expert knowledge and was conducted in accordance with the following rules: for cereals, rapeseed, sunflower, and soya, we removed the last half-month and then added 2 months at the beginning of the harvest stage; for potatoes, sugarbeet, maize, and rice, we removed the last half-month and added 3 months.

#### 2.1.4. Manual Photo Pre-Processing by Visual Assessment

The dataset was then visually assessed with the use of the PyGeon library to remove examples not suited for the study. The protocol consists of selecting a subset of at least 400 photos from the previously identified stack of mature crops photos, loading them into Pygeon, and flagging their suitability of use. Photos were selected on the basis of what one could expect in farmer photos: artificial backgrounds (map, hand, leg, and pivots) and low-quality photos (e.g., against the sun, shadowed, etc.) were not allowed. Close ups showing individual leaves, ears, grains were also removed. Overview photos where the crop appeared somewhere in the background, usually mixed with other elements (a road, neighbouring field, etc.) were also removed. Photos with seeds only on a bare soil background (which mostly occurred for cereals, soy, and maize) and other obviously wrong photos were also eliminated, although this happened only a few times.

Photos flagged as not suitable to train on in this stage were later manually classified into one of the six categories of unfavorable conditions. These were out of season (too early or late in the season), out of protocol (too close or too far away to image the plant matter adequately), or either being blurred or there being a foreign object in the photo. A total of 354 images were selected in this way while making sure that there was at least one photo per year per LUCAS land cover class and per unfavorable condition. An example of unfavorable conditions for common wheat (B11) is shown on Figure 3.

The final clean dataset used in this study is available for download from an FTP server (https://jeodpp.jrc.ec.europa.eu/ftp/jrc-opendata/DRLL/LUCASvision/, accessed on 10 July 2023).

### 2.2. Method

The study makes use of a CNN for an image classification CV exercise with a balanced training and inference set. There are two rounds of training and parallelized inferencing that make up the hyper-parameterization workflow (Figure 4): one without and one with data augmentations (flip, brightness, etc.). After the final augmented inference, the best model was identified, and it was fed with a much larger imbalanced inference set, which was supposed to represent a quasi-operational scenario. Specific and innovative post-processing techniques are also explored.

#### 2.2.1. Sample Selection for Training and Inference Set(s)

The selected number of photos per class for training was set to 400 following the current state of the art [28]. In order to select the set, a stratified sample across NUTS0 regions in the EU was made from the MMEC dataset (from Section 3.1). This was carried out with the idea of having equal representation across EU countries, which allows for articulating conclusions on the European scale.

In order to shorten the processing time, instead of using the entire leftover (post-training-set-selection) set of images for inferencing, this study makes use of a custom inference set sampled out of the leftover set. A total of 85 (the total number of examples of the least represented class (B12)) images per class were selected with a geographic distribution that matched that of the training set. This ‘‘balanced” inference set was used during the first and second stages of inferencing (Figure 4).

The last set of images to be discussed is the ‘‘imbalanced” inference set, which includes all the photos left after the training set selection with all classes capped at 1000 examples per class. This set includes the previous ‘‘balanced” set, which is the one used on the identified best model in order to judge the possibility of using the model as an operational tool. This was also the set that any further developments were tested on.

#### 2.2.2. Hyper-Parameter Search and Best Model Selection

The network used was MobileNet V2. It was selected because it is light, easily trainable, open-source with possibility to transfer learn, and with the option to do on-device inferencing. Furthermore, the achieved results are comparable to those of other state-of-the-art architectures [3]. The images vary in their native resolution (see Appendix A Table A2), but every image in the training and inference set is re-scaled to the net input size of 224 × 224. The effects of this re-scaling are discussed in Section 4.4. The V2 MobileNets are trained for 3000 epochs with the following settable parameters: learning rate, momentum, optimizer, and batch size. These variables were experimented within a random space [29] to generate values for initializing the learning process. In this way, 157 model configurations were tried in order to find the best approximation for solving the problem. The model’s performance was then tested by carrying out an independent inference exercise on the dedicated balanced set. The models were then ranked based on their overall accuracy (OA) to find the top five performers. This completed the first round of training.

For the second round, the top five performers arewerun through another cycle of training with the same configuration but adding image augmentations, in this case, random brightness and horizontal image flips. The same inferencing on the balanced set was carried out to rank the augmented models based on OA. The best-performing of these models was then taken as the overall best model.

#### 2.2.3. Operational Use

After the best model was identified, it was used on the imbalanced inference set (see Section 2.2.1). Because of the class imbalance, it was necessary to use a different metric, the macro-F1 (M-F1) [30]. The results from this inference run are presented in Section 3. It is also on these results that the effectiveness of innovative post-processing techniques were tested and upon which all the discussion was carried out.

#### 2.2.4. Computational Infrastructure

All the code developed for this study is available openly on a git repository (https://github.com/Momut1/LUCASvision, accessed on 10 July 2023). The working environment was carried in a docker image. The processing pipeline was fully reproducible and automated to work by calling shell scripts that carried out the hyper-tuning, inferencing, results derivation, post-processing, and plotting. For more information, the interested reader should consult the *readMe* of the git repository. The processing was carried out on the JRC BDAP, an in-house, cloud-based, versatile, petabyte-scale platform for heavy-duty processing [31]. The offered GPU services work on a NVidia GeForce GTX 1080 Ti with 11 GB memory, CUDA version 10.1, and a CUDA driver version 418.67. Pre-processing, launching, and post-processing were carried out in the JEO-lab layer of the platform in a Jupyter notebook docker container running Tensorflow 1.3.0.

#### 2.2.5. Equivalent Reference Probability Filter

Post-processing the results from ML/DL exercises is an established practice in practically all such workflows [32,33,34]. What it usually consists of is the selected removal, based on some criteria, of a substantial enough number of the incorrectly classified examples in order to increase model performance while simultaneously not falling into the trap of “cherry-picking” one’s results.

In classification problems, analysts can employ a filter on probability, thus keeping only examples for which the network has output an MP of the winning class above a threshold. The analyst then decides where to put the threshold in order to control the rigorousness of the filter; it should be higher for more stringent classification and lower for a more lenient one. The first problem with this is that it depends heavily on the user’s decision and is thus, to a degree, arbitrary. The next problem is that the filter is one-dimensional, so one can only set a threshold along a single axis. Introducing other, or indeed multiple, dimensions to this process would allow for different spreads of the data in the given space. The intuition is that given the chosen dimensions, the data would neatly split between correct and incorrect classifications and allow for more precise filtering. The desired outcome from such filtering would be to remove the biggest amount of incorrectly classified examples without removing too many correctly classified examples.

The proposed method works with a metric based on information theory, the equivalent reference probability (ERP), as described in Bogaert et al. [6]. In information theory, information is the measure of surprise from an event. Rare or low probability events are surprising and hence carry more information, and vice versa (Equation (Equation 1)). Entropy is the information for the probability distribution of the events of a given variable (Equation (Equation 2)). A low entropy means there is a more pronounced difference between the MP for a given class and the rest of the probabilities for the remaining classes. In Bogaert et al. [6], the authors make use of the difference of information (Equation (Equation 3)) between a reference class (preferably the most probable class) and all the other classes in the probability vector. Because $E[D(i∥i^{*})]$ is unrestricted in terms of potential values, and because it needs an upper and lower band in order to be interpreted, the authors suggest using ERP (Equation (Equation 4)). It is a single metric where ERP∈{0,…,1}; values approaching 1 mean a very high confidence in the most probable class with the most equal distribution of the remainder to the other *k* classes.
(1)$$h\left(x\right)=-log_{2}\left(p\left(x\right)\right)$$
(2)$$H\left(X\right)=-\sum_{i=1}^{n}P_{i}log_{2}P_{i}$$
(3)$$E[D(i||i^{*})]=\log p_{i}^{*}-\frac{1}{1-p_{i}^{*}}\sum_{i\setminus i^{*}}p_{i}\log p_{i}$$
(4)$$p^{*}=\frac{exp(E[D(i∥i^{*})])}{exp(E[D(i∥i^{*})])+k-1}$$

The appropriate thresholds for ERP and probability were ascertained with a custom function that iteratively moves the threshold down the line. At each step, it counts the number of disqualified incorrect images while trying to keep the number of correctly classified ones below a certain percent. The settable parameter for this function is thus the percentage of correctly classified examples the analyst is willing to discard. After ascertaining the thresholds, the space within the scatter plot is divided into four quadrants. Through the iterative exclusion of one or combinations thereof of the examples in these quadrants, the analyst can perform more precise filtering on the results.

## 3. Results

Results are divided into five sections. First, we present the MMEC dataset, and second, the best-performing model is presented. Third, the confusion matrix and M-F1 score for the best-performing model are shown alongside the producer (PA) and user (UA) accuracy. Fourth, we show the improvement generated from employing an ERP filter, and lastly, we present the performance of the model when faced with images from unfavorable conditions for each class, simulating the operational use of the model.

### 3.1. Mature Major European Crops

The processing chain from Section 2.1.2 and Section 2.1.3 produces a dataset of 169,460 LUCAS photos of mature crops across 25 EU member states. Utilizing the manual labelling as described in Section 2.1.4, the study also publishes 15,876 high-quality, ready-to-train-on photos, each of which has been manually checked and verified to exhibit a clear view of the crop in its mature, pre-harvest stage with no visual obstructions or foreign objects into the frame. Each class has more than 400 photos, allowing for considerable leeway in training set selection. A summary of the final cleaned dataset across MS and crops is provided in Table 1, where we see the 12 output classes and the number of examples per class. The geographical visualisation of the same information is found in Figure 5.

### 3.2. Best Performing Model

The models were ranked using OA on an independent inference set of 85 images per class (Table 2). The best model was identified as number 78, achieving an OA of 79.4%. The relevant parameter settings are a learning rate of 0.0035148759, a batch size of 1024, and momentum of zero; the optimizer used was gradient descent (GD). The M-F1 and OA on the test set are identical, as we are dealing with a balanced inference set. The last column shows the best model (78) applied over the imbalanced inference set (see Section 2.2.1). The model was exposed to 7722 more examples, and the drop in M-F1 was 0.0369, meaning the model is trained and generalizes very well over larger datasets. While the second- and third-best-performing models had higher training accuracies, their test accuracies were considerably lower. The lowest-performing model (id 139) was evaluated with a test accuracy of 0.702, illustrating the range of performances encountered, with a nearly 0.1 difference between the top and bottom performers.

### 3.3. Confusion Matrix

The confusion matrix for the best model (78) run over the imbalanced operational inference set is presented in Figure 6. It is clear that the majority of confusion happens between the cereal classes (B11–B15) and with grasslands (B55). In fact, the difference between the average PA of all crops, excluding grasslands, and the average PA of the cereal classes is 27.9, and for UA, the difference is 30.9. The class that is most commonly miss-classified as a false positive is durum wheat (B12) with a UA of 10.8; the low score arguably has much to do with the unequal representation of the class. The best-performing class is maize (B16), with a PA of 95.5 and UA of 95, followed closely by rape and turnip rape (B32), showing the clear separation of both from the other classes.

### 3.4. Equivalent Reference Probability Filter

The application of the quadrant filtering method using ERP and MP is shown in Figure 7. The dotted lines represent the thresholds identified by the functions described in Section 2.2.5. The settable parameter is fixed at losing no more than one percent of the correctly classified images, meaning the identified thresholds are the most conservative ones. They are 0.46 for MP and 0.2 for ERP. The inscribed feature shows the number of true and false classifications in each quadrant as labelled by their respective quadrant ID. Although similar, there is a notable difference in the distribution of the true and false classifications, which is visible in the smooth fitted lines for each group.

The results achieved from employing such filtering are presented in Figure 7 in the uppermost right corner. There is an M-F1 increase of 0.6 from not using any filter and of 0.2 from using only the MP filter.

### 3.5. Unfavorable Conditions

The best model (78) was applied over a stratified sample of 1 photo per year, per LUCAS LC1 class, and per unfavorable condition, totalling an inference set of 354, meaning 59 photos per unfavorable condition (see the examples in Figure 3). A boxplot of the Top1 probability for each unfavorable condition is presented in Figure 8. The conditions were compared firstly to a reference set of quality images that were randomly sampled to have the same distribution as the sets of the conditions and secondly to the entire imbalanced inference set. Model 78 is most confused about photos with foreign objects, landscape photos, and photos showing the crop after its harvest period, with blurry, early, and especially close-up photos performing significantly closer to the reference in terms of Top1 probability.

The actual classification results are presented in Table 3. The worst results are achieved with photos exhibiting post-harvest conditions with an OA of 20%, and early crops and examples with a foreign object in the frame, with values of 31% and 37%, respectively. The unfavorable conditions that impact the performance the least are blurry and overly close-up photos (54%). This illustrates that a clear protocol is needed when such automated procedures are used within operational workflows, such as for the CAP [24]. In addition, models can progressively be trained with a set of photos covering a wider range of conditions to improve their generalisation capacity.

## 4. Discussion

### 4.1. Context

Recently, several relevant studies were published. Zheng et al. [18] presented the CropDeep dataset, over which they tested state-of-the-art classification and detection DL algorithms. They achieved an averaged accuracy of 99.81% over the CropDeep datasets. These results are impressive, although not directly comparable, as the images were collected from robots in a sterile greenhouse environment, allowing for image conditions to be identical between acquisitions. They furthermore used average accuracy as a metric over an imbalanced inference set, which is not in accordance with the literature [35]. Gao et al. [36] achieved an accuracy of 99.51% in differentiating 30 wheat cultivars at the flowering (most mature) stage. This is very impressive, considering the present study suffered the most error when trying to discriminate between the various cereal classes. The difference is again in the lab quality of the images taken, whereby each image exhibits a single plant on a white background. d’Andrimont et al. [4] achieved a M-F1 score of 62.3% for 10 classes using street-level images. The current study outperformed the cited work by 13.4%, though this can be attributed to the lower presence of noise in the images fed to the model.

This study presents the first use of the LUCAS cover dataset for automatic crop identification. Indeed, it is the first study to apply DL for crop identification on still images that were not taken in a controlled environment and that come from a wide variety of sensors, which truly mimics an operational scenario. Secondly, the study produces an automated method to attach crop life-cycle stage information to a database of photos. Third, the introduction of quadrant filtering is a step towards a new state of the art for more precise post-processing filtering. Whether using crop calendars to extract photos for specific crop life-cycle stages, or using the dataset as a whole, the authors believe that various lines of research may be developed using the LUCAS cover photos.

### 4.2. ERP Filtering

A main achievement of this study is the exploration of methods for filtering classification results to achieve better performance and to quantify uncertainty. This study made use of ERP as a metric for assessing this uncertainty. According to the literature, ERP has been shown to be more robust than MP in classifying pixel-level thematic uncertainty [6], more precise than majority voting in post-processing speckle removal of classified maps [37], and more flexible than OA in terms of the independence of the distribution of the validation data [38].

In practice, MP and ERP are connected, which is clearly visible in the distribution of both groups (correct and incorrect) in the space where the joint probability reference distribution is not null in Figure 7. From the marginal distribution plots, we can see that this connection is inverted; there is a high peak in the low values of MP for the incorrectly classified points and a high peak in the high values of ERP for the correctly classified points. Furthermore, as shown in Figure 9, ERP performs significantly better than MP in post-processing filtering. Because ERP and MP are both probabilities that are in the range between zero and one, their direct comparison in this regard is straightforward. Firstly, in subplot A, for an equal threshold value, the M-F1 value is always higher when utilising ERP over MP. This means that ERP is a much better estimator of uncertainty and manages to capture to a finer degree the nuances that distinguish an incorrect from a correct classification. It needs to be mentioned that this is partly due to the fact that, while for MP, the smallest possible threshold value is relatively high (0.20), with ERP, it is found at the first stage of filtering (0.01). As seen on the secondary Y axis, which shows the number of images left in the set after performing the filter, this process is not without cost; the number is, for every threshold value, less for ERP than for MP. Nevertheless, it is always preferable to have a larger spread of the data over which to set thresholds, which is especially true when the analysis needs to be conservative regarding the number of correct classifications it is willing to lose.

Furthermore, the histograms in subplots B and C show the points at which the proportions of correct and incorrect classifications for each threshold value, represented by the height of the gray and yellow bars, relative to the red bar, change in favour of the correct ones. While with MP this point arrives at 0.54, for ERP, the change is already present at 0.24. Hence, the relative cost in terms of number of examples disqualified due to the threshold setting is proportionately lower with ERP in order to achieve the same increase in M-F1.

### 4.3. Limitations

Although several novel aspects have been highlighted, some limitations are present in our study. Firstly, there are issues with the pre-processing of the data, including, in particular, the fact that CC information comes from a variety of sources. Although these are official CCs, which have been harmonized, the fact that the study treats them as a priori semantically harmonized could be problematic. Because organizations, based on their goals, have different data collection, processing, and publishing protocols, it is conceivable that the data were intended for a different use. For example, some CCs might be designed to track crop phenology, to help manage labour or funds, or to track fertilizer, pesticide application, or other agricultural practices. Because data on specific crops are scarce at the EU scale, all CCs were treated as phenology-relevant. This issue becomes even more pronounced when considering the expert knowledge and model output gap filling. Indeed, the concern that the latter would introduce error into the results was such that the study went ahead and calculated the M-F1 for each country (NUTS0 region) for which the crop calendar information was derived from expert knowledge or model output, and this was then compared to the reference M-F1. No clear drop in M-F1 based on the origin of the mature crop information was registered from this analysis.

Another data issue is that bias can be introduced during manual selection by visual assessment. Other than errors due to distraction during annotation, the annotator undoubtedly bases their decision on which images to keep and discard based on their own discretion. For example, the annotator had to consider questions whether there should be any sky or abundance of soil visible on the image; if the crop on a given image could be considered mature enough; and, especially for the cereal classes, if a given label was correct. The matter is even more pronounced when selecting examples for unfavorable conditions, during which, for example, the distinction between “blurry” and “close” was sometimes hard to make. Often, the object in the “object” class and the visual appearance of the landscape in the “landscape” class were very varied, and sometimes the image showed more than a single unfavorable condition, such that the crop can be both early in the season and blurred out, in which case, one could have used multi-tags. Such issues were considered prior to undertaking each task, yet the possibility of bias has to be mentioned.

Secondly, there are issues related to the processing logic of certain steps. One such issue is the identification of threshold points for MP and ERP to generate quadrants. The way the custom function works is by peeking into the correct–incorrect classification results in order to iteratively arrive at the threshold, with the main consideration being keeping the number of disqualified correct classifications below a certain percentage. In a sense, this means putting the proverbial data cart before the horse, as instead of simply using the values of whichever chosen metrics, the function also considers the result of the classification.

### 4.4. Recommendations

There are several recommendations that would be a logical continuation of the work. In terms of class selection, the major part of the confusion stems from the cereal classes (Section 3). This makes sense, as to distinguish between them can sometimes be troublesome even for a skilled professional. When cereals are treated as a grouped cereal class, they are easily set apart from the rest of the crops, but when assessing the differences between the cereals, the structure of the fruit, stem, and leaf organs can look too similar. Indeed, the approach in Gao et al. [36] yields such good results exactly because the model is designed to pick up on the subtle differences between the varieties. In the present case, grouping the cereals together would produce a M-F1 of 88.2 without and 90.4 with quadrant filtering, which is 12.5 and 14.7 points higher than the achieved result. Ideally, one could capture the cereal class first at these higher ranges of M-F1 and then have a separate model that deals solely with classifying the type of cereal, variety, or cultivar.

Concerning the point of being more robust in identifying thresholds, or more generally on the topic of splitting the space in Figure 7, one could build a kind of Bayesian discriminant rule in order to generalize the combination of the two 1-D thresholds to a 2-D threshold. This can be executed by taking into consideration the joint distributions and would yield a single curve that separates correctly and incorrectly classified examples.

An always-current topic in DL for CV is the effect of resolution on the results. In this case, one can discuss both the input resolution of the source images and the input resolution of the net in use. Firstly, the range of values of images’ resolutions in the inference set vary from 480 to 3504 in height and 640 to 4672 in width—a 7.3 times difference in each dimension. Almost 65% of the images are of a resolution of 1600 × 1200, with another 22% being 2048 × 1563 (for a full breakdown of available image resolutions, check Appendix A Table A2). With such a spread, one can imagine that the level of detail visible on images from either end of the range is quite different. When measuring the correlation between image resolution and the proportion of correctly classified examples for each resolution bin (Figure A2), the study found an R-squared value of 0.009, meaning the correlation for this set of LUCAS photos is almost none. Secondly, the net input size is 224 × 224, meaning each parallelogram image of the training and inference set is re-scaled to this square size. Intuitively, one can say that larger images would lose more information during re-scaling than smaller ones. In reality, the re-scaling turns the problem into a detection of the major structural features of the crops (e.g., broad leaf vs. cereals, colouring, having recognisable flowers or not), where resolution does not matter as much. This would also shed light as to the reason why the network has trouble distinguishing between cereal classes. The analysis still serves to illustrate that the method is developed to handle images from different resolutions equally well. This further showcases the policy relevance of the work, as in an operational context, a regulating body is expected to receive evidence images in a variety of image resolutions.

## 5. Conclusions

This study provides a subset of LUCAS Cover photos for 12 major crops across the EU to deploy, benchmark, and identify the best configuration of MobileNet for the classification task, to showcase the possibility of using entropy-based metrics for the post-processing of results, and finally, to show the applications and limitations of the model in a practical and policy-relevant context. This work has produced a dataset of 169,460 images of mature crops for 12 classes, out of which 15,876 were manually selected as representing a clean sample without any foreign objects or unfavorable conditions. The model that performed best in identifying crops achieved a macro F1 (M-F1) of 0.75 on an imbalanced test dataset of 8642 photos. Using metrics from information theory resulted in achieving an increase of 6%. The most unfavorable conditions for taking such images, across all crop classes, were found to be too early or late in the season. The proposed methodology shows the possibility of using minimal auxiliary data outside the images themselves in order to achieve an M-F1 of 0.817 for labelling 12 major European crops.

## Figures and Tables

**Figure 1 sensors-23-06298-f001:**
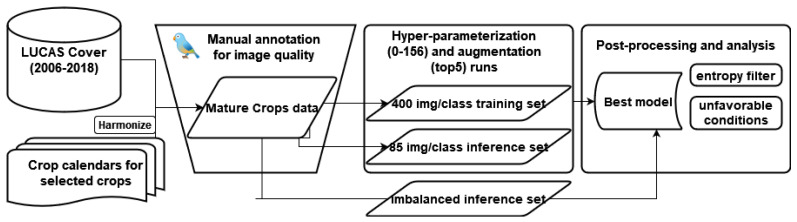
Conceptual diagram of the study. The used data are shown on the left. LUCAS attributes are fused with harmonized crop calendars for the selected crops, after which the combined dataset undergoes a process of manual annotation using the pyGeon library. After annotating enough images of sufficiently high quality, a stratified sample across EU countries is chosen to select the training and inference sets, followed by the DL paradigm (described further in Section 2.2). The DL workflow produces the best-parameterized model, which, in turn, is used to make inferences over a large imbalanced set, where post-processing and further operational context work takes place.

**Figure 2 sensors-23-06298-f002:**
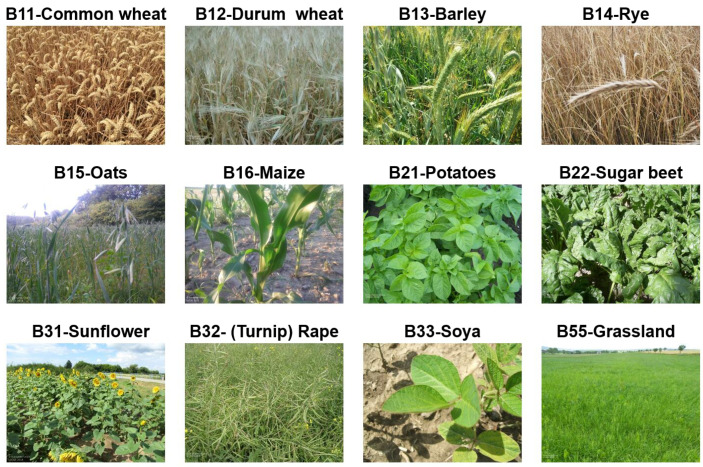
Example of one image per class for the 12 selected major European crops.

**Figure 3 sensors-23-06298-f003:**
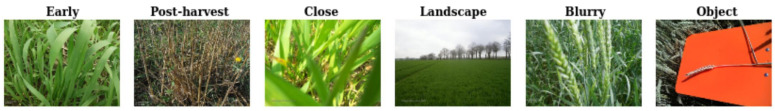
The six classes of unfavorable conditions for common wheat (B11).

**Figure 4 sensors-23-06298-f004:**
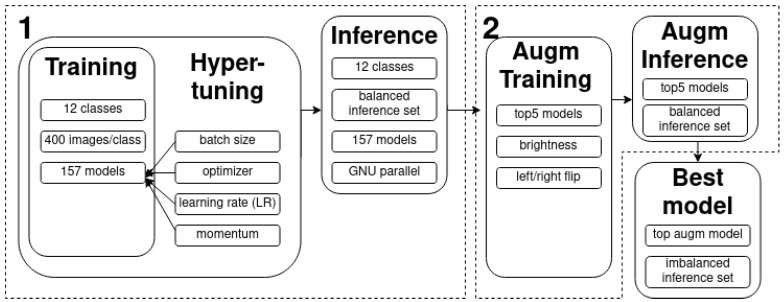
Deep learning processing chain. The numbers in the top left corners of the black contoured boxes indicate the two rounds of training and inferencing–without and with image augmentations.

**Figure 5 sensors-23-06298-f005:**
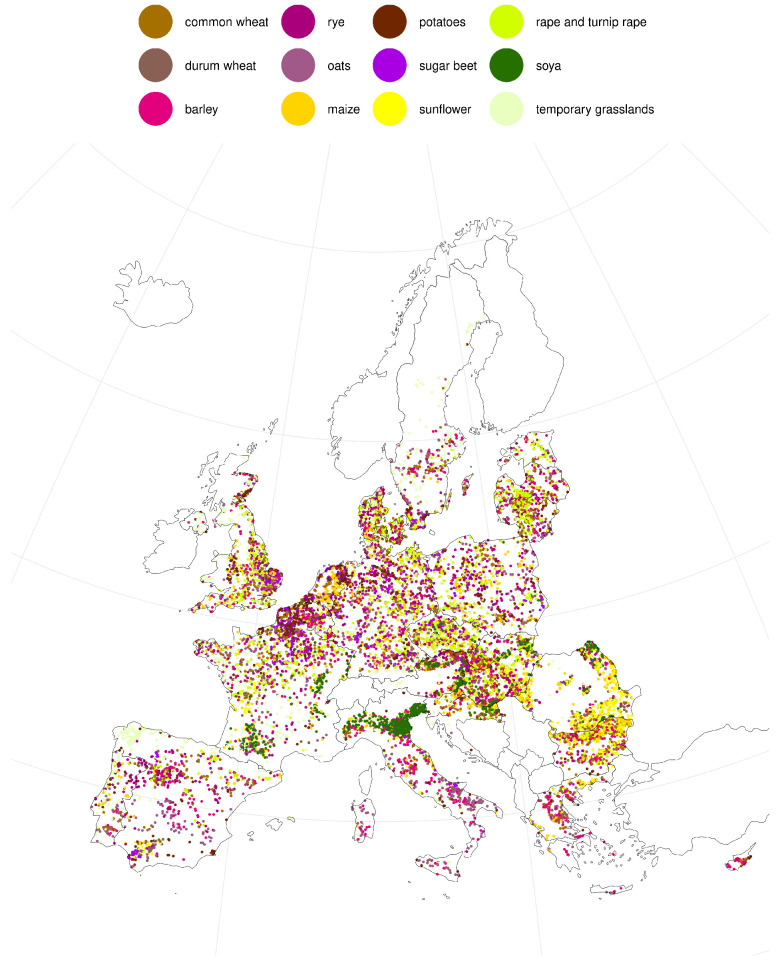
Geographical distribution of 15,876 LUCAS Cover photos across the EU, which have been manually screened and validated as ready-to-train-on. Map projection EPSG:3035.

**Figure 6 sensors-23-06298-f006:**
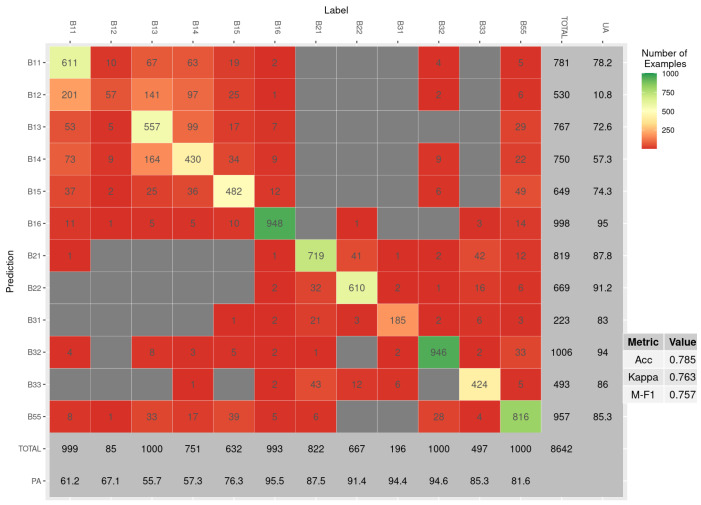
Confusion matrix for best model (78) over imbalanced inference set. The outermost marginal rows show the user and producer accuracy (UA and PA), or precision and recall, respectively.

**Figure 7 sensors-23-06298-f007:**
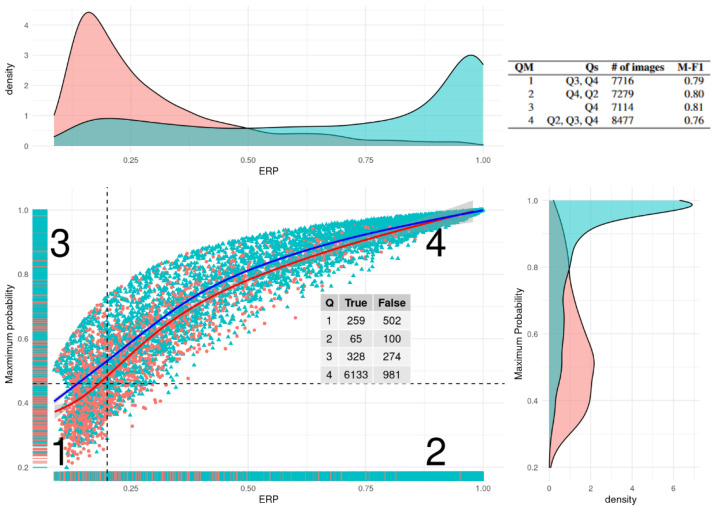
Scatter plot of ERP and probability quadrant filtering with marginal density plots for each variable. In red are the incorrect classifications, and in blue, the correct classifications are shown. Numbers within quadrants indicate the quadrant ID. The data are fitted with smooth lines for correct and incorrect classifications. The uppermost corner shows the results from the quadrant filtering. In order, the columns represent the quadrant method ID, the quadrants included in the method, the number of images, and the M-F1 achieved through the inclusion of the respective Qs. In order, the QMs represent the following: 1. MP only, 2. ERP only, 3. both above their respective thresholds, and 4. at least one above its threshold.

**Figure 8 sensors-23-06298-f008:**
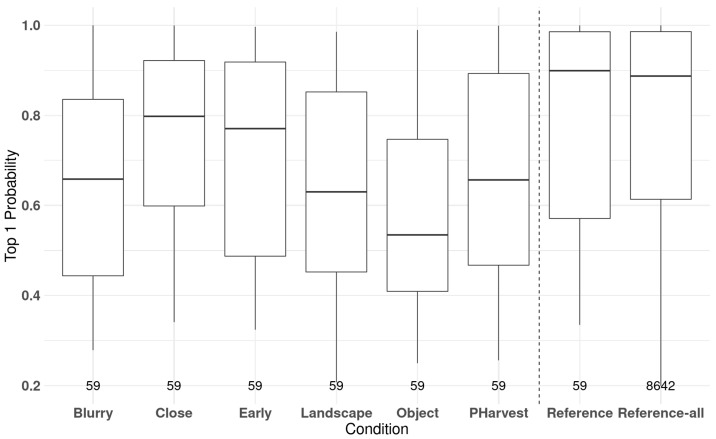
Top1 probability for all examples for a given unfavorable condition with reference to random sample of the same size of the balanced inference set and to the entire imbalanced inference set. Number of examples in each box is given above the condition label.

**Figure 9 sensors-23-06298-f009:**
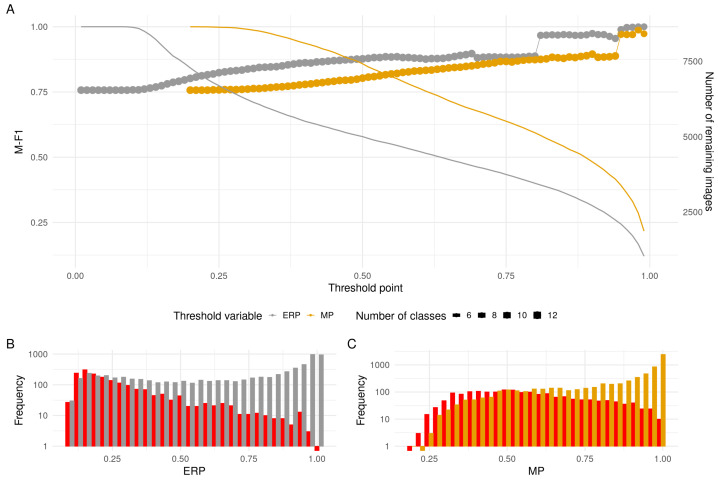
Comparison between using MP and ERP for applying filtering on results. Plot (**A**)’s first Y axis shows the evolution in M-F1 in connected points when applying a threshold on the inference set, and its secondary Y axis shows the diminishing number of photos in continuous planes when applying the same thresholds. Plots (**B**,**C**) represent the distributions on a logarithmic scale (base 10) of values present in the inference set in terms of ERP and MP, respectively. The red bars in both histograms are the incorrect classifications.

**Table 1 sensors-23-06298-t001:** All visually inspected MMEC photos labelled as good to train on from the manual annotation using PyGeon. The marginal rows labelled MMEC show the total number of photos that show mature crops, which have not been visually inspected. The formats of the data are the images themselves, alongside a table with the crop labels and all collected LUCAS information.

	B11	B12	B13	B14	B15	B16	B21	B22	B31	B32	B33	B55	Total	Total MMEC
**AT**	136	32	139	103	29	69	48	67	18	85	122	124	972	3595
**BE**	59	2	71	4	3	39	93	49	0	23	0	62	405	2127
**BG**	179	5	129	19	12	148	11	0	110	90	4	3	710	2855
**CY**	15	1	29	0	2	0	6	0	0	0	0	0	53	207
**CZ**	72	4	28	32	20	47	30	45	4	194	12	55	543	6691
**DE**	156	40	157	176	133	114	177	152	17	220	6	212	1560	24,055
**DK**	132	2	93	105	27	55	21	24	0	184	0	175	818	3226
**EE**	20	0	16	7	8	1	5	0	0	26	0	14	97	612
**EL**	62	81	88	0	25	62	6	7	22	9	0	4	366	1386
**ES**	68	58	85	105	80	22	59	58	34	50	0	178	797	19,582
**FR**	121	97	116	130	118	82	139	143	75	186	142	193	1542	40,989
**HR**	38	0	22	4	7	53	6	2	16	9	34	16	207	434
**HU**	74	34	103	66	30	53	17	8	49	135	49	8	626	7354
**IT**	136	54	175	17	130	54	50	98	17	21	378	166	1296	13,387
**LT**	82	6	65	72	50	1	28	4	0	147	0	31	486	2313
**LU**	3	0	7	2	0	1	0	0	0	1	0	6	20	149
**LV**	66	3	72	44	33	9	12	1	0	110	0	42	392	1763
**NL**	150	0	53	23	2	134	200	115	0	3	0	54	734	1805
**PL**	66	12	40	105	61	98	89	93	1	173	7	67	812	20,542
**PT**	28	7	24	40	50	22	19	0	1	0	0	68	259	877
**RO**	71	28	39	13	25	159	22	16	189	51	88	32	733	3649
**SE**	89	0	67	47	85	11	34	55	0	95	0	167	650	2742
**SI**	29	6	26	2	2	30	6	0	1	5	0	27	134	364
**SK**	89	13	95	27	16	51	18	25	42	164	55	18	613	3091
**UK**	155	0	134	8	84	78	126	105	0	182	0	179	1051	5665
**Total**	2096	485	1873	1151	1032	1393	1222	1067	596	2163	897	1901	15,876	-
**Total MMEC**	47,143	8062	31,500	7296	6582	32,175	4113	4414	6830	13,958	1603	5784	-	169,460

**Table 2 sensors-23-06298-t002:** Output for the three highest-performing models with augmentations plus the output from the best model ran on the imbalanced set. The applied augmentation were left–right flip and random brightness. Shown (in order) are the model number, along with the relevant configuration (learning rate, batch size, momentum, and optimizer), the number of labelled images, the training and validation accuracy, and the M-F1.

Ranking	1	2	3	Best
Model	78	88	4	78
Level	Augm	Augm	Augm	Best Model
LR	0.0035	0.0073	0.0096	0.0035
BS	1024	512	512	1024
Momentum	0	0	0	0
Optimizer	GD	GD	GD	GD
Number of Images	1020	1020	1020	8642
Validation Accuracy	0.7768	0.7789	0.7747	0.7768
Training Accuracy	0.8945	0.8965	0.9238	0.8945
Test Accuracy	0.7941	0.7775	0.7755	0.7854
M-F1	0.7941	0.7775	0.7755	0.7572

**Table 3 sensors-23-06298-t003:** Number of true and false classifications and overall accuracy for each unfavorable condition.

	False	True	OA
Blurry	27	32	0.54
Close	27	32	0.54
Early	41	18	0.31
Landscape	35	24	0.41
Object	37	22	0.37
Post-harvest	47	12	0.20

## Data Availability

Data—https://jeodpp.jrc.ec.europa.eu/ftp/jrc-opendata/DRLL/LUCASvision/, accessed on 10 July 2023. Code—https://github.com/Momut1/LUCASvision, accessed on 10 July 2023.

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
