# Peer review of "Crop Identification Using Deep Learning on LUCAS Crop Cover Photos"

_sensors, 2023, doi:10.3390/s23146298_

Round 1

Reviewer 1 Report

In this manuscript, the main objective is to enhance crop identification using deep learning techniques on a diverse dataset of real-world images. The manuscript's strengths lie in its comprehensive analysis. This study offers valuable insights into deep learning applications for crop identification and provides a foundation for further advancements in the field.

There are several main issues with this manuscript:

1.     The introduction section is excessively lengthy and includes numerous technical details and references. This may make it difficult for readers to grasp the core content of the research at first glance.

2.     Table 1 requires formatting adjustments.

3.     In the data section, it is mentioned that crop calendar data was collected from various sources, but the reliability of the data sources is not addressed.

4.     Manual photo preprocessing through visual assessment is mentioned, but no detailed process is provided regarding how it was performed.

5.     In 2.2.2, it is stated that MobileNet V2 was used as the neural network, but no specific justification is given for this choice.

6.     In 2.2.2, the use of (OA) and Macro-F1 for evaluating model performance is mentioned. Relying solely on these two metrics may not provide a comprehensive assessment of the model's performance across different categories and conditions.

7.     In 3.2, there is a lack of detailed discussion regarding the performance of other models and the selection of parameters. A more comprehensive comparison and analysis would help readers understand why this particular model was chosen as the best-performing one.

In summary, there are several areas in this manuscript that require adjustments. Please carefully revise the manuscript.

Minor editing of English language required

Reviewer 2 Report

Overall, this is a well written manuscript. Authors have provided comprehensive explanation on the problem statements, contributions, proposed methodology and experimental results. There are only some minor comments to be addressed by authors before this paper can be accepted for publications:

1. Conceptual diagram presented in Fig. 1 is too small and can be enlarged. 

2. Please provide a summary of the final clean dataset used in this study such as the numbers of output classes, type of output class, the numbers of dataset for each output class and etc.

3. What are the optimizer used to train the MobileNet V2?

4. The git repository provided is not accessible.

5. Other performance metrics such as specificity, precision, F1 score should be considered in the performance analyses.

6. Confusion matrix in Fig. 6 needs to be enlarged.
